## Perspective

eSource implementation; oncology clinical trials; EHR-to-EDC data integration; clinical trial digitization; research workflow optimization

**Corresponding author:**
Mats Sundgren;
Email: mats.sundgren@i-hd.eu

# Accelerating eSource scale-up in oncology clinical trials: The i~HD Task Force initiative

Christophe Maes[1], Dipak Kalra[2], Tracy Acito[3], Nadir Ammour[4], Paul Basset[4], Sarah Burge[5], Peter Castleyn[2], Ross Caldow[3], Camille Couvert[4], Amy Cramer[6], Chris Harrison[7], Joeri Holtzem[6], Pavitra Mariappan[7], Paul Jacobs[3], Lars Fransson[8], Veronique Berthou[4], Laurice Jackson[9], Nancy Wetzel[9], Christopher Thompson[9], Sharon Klein[9], Robert Green[6], Fakhry Kaoukdji[8], Michael Ward[9], Felix Nensa[10], Joe Lengfellner[11], Anna Patruno[11], Dawn Snow[4], Isabel Virchow[10], Angela Fritsche[12], Pascal Coorevits[1] and Mats Sundgren[2] ![ORCID]

[1]Faculty of Medicine and Health Sciences, Department of Public, Ghent University, Ghent, Belgium; [2]European Institute for Innovation through Health Data, Ghent, Belgium; [3]Regeneron Pharmaceuticals, Terrytown, NY, USA; [4]Sanofi Recherche and Développement, Paris, France; [5]Cambridge University Hospitals NHS Foundation Trust, Cambridge, UK; [6]Johnson & Johnson, Beerse, Belgium; [7]AstraZeneca One MedImmune Way Gaithersburg, MD, USA; [8]AstraZeneca R&D Gothenburg, Sweden; [9]Eli Lilly & Company Lilly Corporate Center, Indianapolis, IN, USA; [10]Institute for Artificial Intelligence in Medicine (IKIM), University Hospital Essen, Essen, Germany; [11]Memorial Sloan Kettering Cancer Center, New York, NY, USA and [12]Mayo Clinic Comprehensive Cancer Center Clinical Trials Office, Rochester, MN, USA

## Abstract

eSource – particularly EHR-to-EDC – is an emerging paradigm in clinical research that enables automated transfer of electronic health record (EHR) data into electronic data capture (EDC) systems, with the potential to reduce site burden, improve data quality and accelerate oncology clinical trial workflows. However, widespread implementation remains limited due to technical, regulatory and operational barriers. To address these challenges, the European Institute for Innovation through Health Data (i~HD) launched the eSource Scale-Up Task Force in 2024. This multi-stakeholder initiative brings together leading oncology centres and pharmaceutical sponsors to establish a consensus-driven roadmap for eSource adoption. Central to this effort are three foundational resources: readiness criteria for early adopters, a performance indicator framework for monitoring success and an operational playbook to guide implementation. This article provides a structured overview of the Task Force's objectives, collaborative model and outputs, with specific attention to its focus on interoperability, regulatory alignment and real-world validation. While initially developed for oncology, the Task Force's framework is applicable across therapeutic areas characterized by data-intensive workflows.

## Impact statement

Manual data workflows remain one of the most persistent barriers to efficient, high-quality oncology research – consuming staff time, introducing errors and delaying patient access to innovative treatments. This article addresses those systemic inefficiencies through a roadmap for eSource technology adoption, co-created by a cross-industry consortium of leading hospitals and pharmaceutical sponsors.

The framework outlined here enables hospitals and sponsors to automate EHR-to-EDC data transfer, reducing site burden and improving data integrity. More importantly, it facilitates faster, safer and more inclusive trials – laying the foundation for precision medicine studies that rely on complex genomic, imaging, and real-world datasets. By aligning with global regulatory expectations and providing practical tools validated by early adopters, this initiative empowers research centres to implement eSource sustainably and at scale.

The broader value of this work lies in its transferability across therapeutic areas – extending beyond oncology to fields such as neurology and rare diseases. It illustrates how collaborative governance, implementation science and interoperability standards can converge to bridge the longstanding divide between clinical care and research. This shift is vital not only for improving trial efficiency but also for ensuring that diverse patient populations gain faster and more equitable access to cutting-edge therapies.

## Introduction: rising complexity

Oncology clinical trials are entering an era of profound transformation, driven by the exponential growth of clinical data, increasingly stringent regulatory requirements and the persistent inefficiencies of manual data handling. Traditional trial workflows require research teams to extract, transcribe and validate patient data from EHRs into EDC systems – a duplicative process that affects more than half of all trial data elements and demands extensive verification, often consuming substantial operational resources (Coulter, 2023; Hamidi et al., 2024). As precision medicine accelerates the inclusion of genomic, imaging, biomarker and real-world data in cancer studies, the sheer volume and complexity of data per patient have become overwhelming for research sites and sponsors (Sundgren et al., 2024).

The adoption of eSource technology – particularly EHR-to-EDC data integration – has emerged as a transformative solution in clinical research (Cramer et al., 2024) (Figure 1). The term *eSource*, as used by regulators such as the Food and Drug Administration (FDA) and the European Medicines Agency (EMA) and widely adopted across the pharmaceutical industry, refers to the direct, automated and regulatory-compliant transfer of data from hospital EHR systems to clinical study databases. This approach significantly reduces site burden, improves data accuracy and accelerates trial timelines. Built on interoperability standards such as HL7® FHIR® (Fast Healthcare Interoperability Resources) and SMART on FHIR APIs, eSource also enables structured and, increasingly, AI-assisted extraction of clinical data (Chopra et al., 2023; Nashwan and Hani, 2023; Chakrabarty and Mahajan, 2024), including from unstructured sources such as clinician notes, pathology reports and radiology narratives.

Despite its potential, the adoption of eSource across the industry remains fragmented. Persistent barriers include a lack of system interoperability, evolving regulatory interpretations and varying degrees of site technical and organizational readiness. Many institutions face challenges aligning their EHR systems with sponsor EDC platforms, while others struggle with limited staffing or training to implement new workflows. As a result, scalable, industry-wide transformation has proven difficult and a unified framework for implementation has been lacking.

To address these barriers, the European Institute for Innovation through Health Data (i~HD) launched the eSource Scale-Up Task Force in Q1 2024. This impartial, cross-industry consortium unites healthcare providers, pharma sponsors and regulatory experts in a shared mission to accelerate eSource adoption across oncology research centres.

In this Perspective article, we outline the collaborative governance and implementation roadmap developed by the i~HD Task Force to guide scalable adoption of eSource trials. We present how the Task Force is bridging the gap between innovation and implementation by synthesizing insights from pilot programs, institutional leaders and implementation science. The article introduces a strategic roadmap anchored by practical tools and real-world validation to support eSource scale-up in oncology – and eventually across other data-intensive therapeutic areas. It builds on empirical findings from six large oncology centres, which documented significant operational improvements, including a 99% reduction in transcription errors and over 50% reduction in site burden (Sundgren et al., 2025).

## The challenge: redundant workflows and data burden

The operational demands of oncology clinical trials are rapidly intensifying, driven by the evolution of personalized medicine, real-world data and novel biomarkers. As the volume and complexity of clinical data grow, so too do the burdens placed on research sites and sponsors. At the heart of this issue lies a fundamental inefficiency: the manual extraction, transcription and validation of data from EHRs into EDC systems. This duplication affects more than half of all trial data and consumes significant resources that could otherwise be redirected toward scientific advancement and patient benefit.

**Increasing data complexity in modern oncology trials**. Today's oncology trials are characterized by their data intensity. The rise of

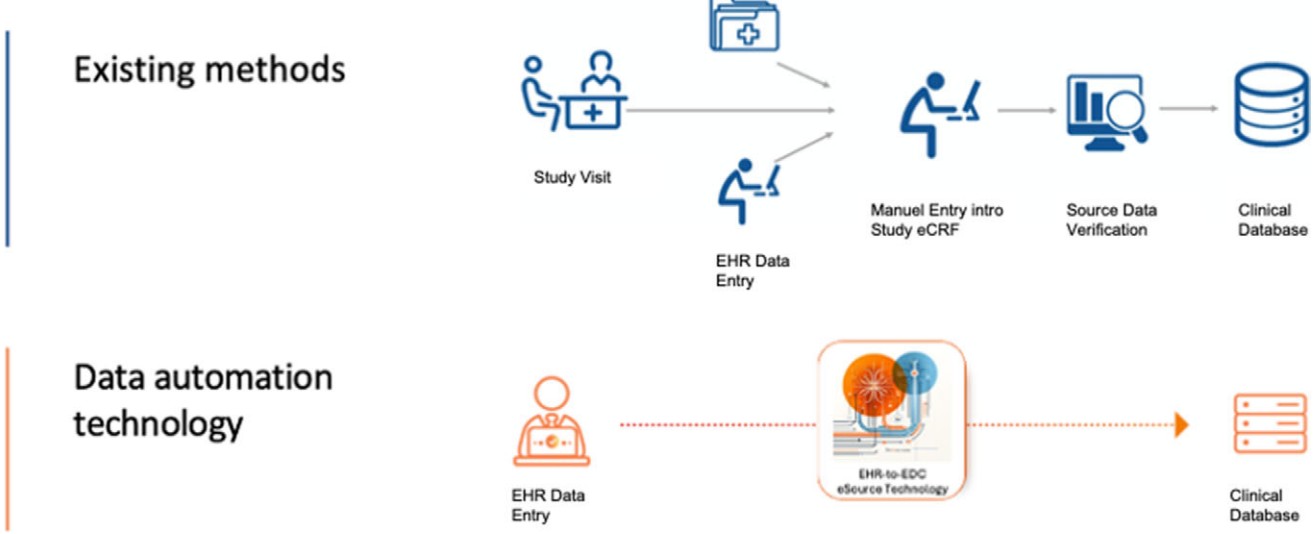

**Figure 1.** eSource technology vs existing methods.

precision oncology has led to the routine integration of complex datasets such as genomic profiles, radiological imaging and patient-reported outcomes alongside standard clinical data. In modern oncology studies, the data burden per patient has expanded dramatically. For example, Phase I oncology protocols now collect over 27,000 data points per patient – more than six times the average in non-oncology trials – with Phase III oncology studies collecting more than twice as many data points as their non-oncology counterparts (Tufts CSDD, 2022). Each of these data elements must be documented, reviewed and often verified manually.

This growing complexity is exacerbated by siloed health IT systems, non-standardized documentation and the use of unstructured data. Clinical notes, radiology reports and pathology results are frequently embedded in free-text formats, requiring human interpretation and re-entry into EDC systems. This not only prolongs data entry, but increases the potential for transcription errors, inconsistencies and missing information – all of which threaten data quality and regulatory compliance.

**The burden of redundant data entry.** Clinical research coordinators (CRCs), who play a critical role in trial execution, carry the weight of redundant data handling. Multiple studies, each with distinct case report forms (CRFs) and timelines, place extraordinary pressure on site personnel. As highlighted in recent interviews with CRCs at Memorial Sloan Kettering Cancer Center and other eSource Champion sites, manual data entry is not just time-consuming – it detracts from high-value activities like patient engagement, protocol adherence and real-time data monitoring (the opportunity.

The burden extends beyond staff time. Redundant workflows necessitate extensive source data verification (SDV) by sponsors and contract research organizations (CROs), further inflating timelines and costs (Hamidi et al., 2024). Estimates from EHR-to-EDC initiatives suggest an average of five minutes per data point is required for manual entry and verification in oncology trials – translating to many thousands of hours per study. In addition to transcription itself, manual data entry generates substantial downstream workload, including query resolution, data reconciliation and extensive data review cycles (Ehidiamen and Oladapo, 2024) – activities that are estimated to account for up to 25%–40% of total data management costs in oncology trials (Hamidi et al., 2024).

**Compliance and data integrity challenges**. Regulatory agencies worldwide have emphasized the importance of high-quality, traceable clinical data. Standards such as ALCOA+ (Attributable, Legible, Contemporaneous, Original, Accurate,and Complete) and compliance frameworks like the Global Data Protection Regulation (GDPR) and the Health Insurance Portability and Accountability Act (HIPAA) underscore the necessity of data that is not only accurate but also secured and auditable (Ehidiamen and Oladapo, 2024). However, when data is manually transcribed from EHRs into sponsor systems, the audit trail can become fragmented. Ensuring contemporaneous and original data becomes significantly more difficult, especially when EHR systems are not designed to support regulatory-grade documentation workflows. Furthermore, discrepancies between EHRs and EDC entries introduce potential risks during inspections and audits. The lack of integration between clinical care and research systems often leads to conflicting information, data reconciliation delays and regulatory red flags.

**Impact on timelines, cost and trial viability**. Delays caused by data collection and verification bottlenecks can significantly impact a trial's critical path. Time lost to manual workflows translates to slower patient enrolment, delayed database lock and prolonged time to regulatory submission. For sponsors in competitive oncology indications, such delays can result in missed market opportunities and reduced investor confidence. For patients, lengthy timelines increase the risk for patient safety identification and limited access to breakthrough therapies. From a financial standpoint, the inefficiencies of current processes scale dramatically with trial size. In multi-centre, global oncology trials, duplicated processes across dozens or even hundreds of sites can add millions in operational costs – resources that could otherwise support trial expansion, diversity initiatives, or exploratory endpoints. Compounding these challenges, recent NIH funding cuts in the U.S. – including reductions in indirect cost reimbursements from 60% to 15% – have significantly constrained research infrastructure at cancer centers, further amplifying the need for more efficient, digitally supported trial operations (Busiek, 2025; Rhodes, 2025).

**Fragmented stakeholder ecosystem.** Lastly, the complexity of oncology trials is compounded by a fragmented ecosystem of stakeholders. Clinical sites, sponsors, CROs, EHR vendors, EDC providers and regulatory bodies often operate with misaligned goals and disconnected systems. Without a unified framework for interoperability, data flow remains linear, manual and error-prone. Attempts to streamline data collection often stall at the site level due to a lack of technical infrastructure, unclear regulatory guidance, or insufficient support from sponsors. As a result, even when sites are willing to adopt innovative solutions like eSource, they may lack the resources or organizational mandate to implement them effectively (Cramer et al., 2024).

## The opportunity: eSource and EHR-to-EDC integration

Amid the rising complexity of oncology clinical trials, the adoption of eSource technology – particularly EHR to EDC integration – has emerged as a practical, scalable and transformative solution. eSource enables the direct, automated transfer of clinical data from hospital EHR systems into study databases, eliminating redundant manual data transcription, minimizing errors and accelerating timelines. More than a technical upgrade, eSource represents a paradigm shift in how clinical data is collected, managed and validated across the research ecosystem.

At its core, eSource leverages modern interoperability standards, such as HL7®, FHIR® and SMART on FHIR APIs, to securely map and transfer structured clinical data from source systems into EDC platforms. These standards allow seamless connectivity between disparate health IT environments, ensuring that data remains consistent, traceable and compliant with regulatory expectations. Importantly, eSource is not a replacement for research staff, but rather a tool that enhances efficiency and liberates site personnel from low-value, repetitive tasks.

The advantages of EHR-to-EDC integration are compelling. Automation significantly reduces the risk of human error associated with manual entry, while simultaneously ensuring higher data quality and completeness. Regulatory agencies such as the FDA, EMA and MHRA (Medicines and Healthcare products Regulatory Agency) have increasingly supported eSource approaches, recognizing their ability to uphold ALCOA+ principles and streamline clinical trial oversight. With proper validation, eSource implementations can also reduce the burden of SDV by sponsors and CROs, offering direct cost and time savings.

The eSource transfer process involves moving structured EHR data into EDC systems or sponsor databases for clinical studies, including but not limited to randomized clinical trials (RCTs). This

data typically includes laboratory results, vital signs, medications, diagnoses and demographics, standardized through coding systems like ICD, SNOMED CT and LOINC. The process is governed by rigorous quality assurance, patient consent and adherence to global eSource regulatory guidelines. Importantly, eSource enhances existing workflows without replacing manual entry in scenarios where human validation or interpretation remains necessary.

Case studies from early adopters underscore these benefits. At Mayo Clinic, for example, the deployment of eSource tools has allowed staff to focus more on patient interaction and trial coordination, rather than manual transcription. At City of Hope, transcription errors have been virtually eliminated and average data entry time per subject has decreased from 15 minutes to under 5 minutes. These real-world gains highlight the scalability and reproducibility of eSource when implemented within a structured and collaborative framework (Sundgren et al., 2025).

Technological innovation is further expanding the capabilities of eSource. Artificial intelligence (AI) and machine learning (ML) (Adamson et al., 2023) are increasingly applied to extract meaning from unstructured clinical data, such as radiology reports, clinician notes and pathology narratives. This is particularly relevant in oncology, where critical information often resides in free-text formats that are difficult to standardize manually.

Despite these promising advances, eSource adoption at scale has been limited by stakeholder fragmentation, inconsistent technical infrastructure, and unclear implementation pathways. Although pilot initiatives such as EHR2EDC (EIT Health), TransCelerate's eSource project and the FDA's Real-World Evidence guidance have demonstrated feasibility, the field still lacks a unified, cross-industry roadmap (Claerhout et al., 2019; Ammo et al., 2023; Mueller et al., 2023). Common challenges include harmonizing data formats, ensuring interoperability across vendor platforms, securing regulatory confidence and articulating the return on investment for long-term adoption.

In summary, eSource and EHR-to-EDC integration offer a future-ready solution to the operational challenges of modern oncology trials. By improving data quality, reducing administrative burden and accelerating research timelines, this approach has the potential to transform how clinical trials are conducted. Realizing this potential requires not only technological readiness but also the coordinated engagement of stakeholders across the research ecosystem.

## A strategic response: the i~HD eSource Scale-Up Task Force

Based on the experience of the core team members of the task force, supported by numerous papers and publications referenced in this article, the promise of eSource to streamline clinical trials and reduce burdens for research sites and sponsors is supported by growing evidence from early adopters. Yet, despite its potential, large-scale adoption remains elusive – hindered by fragmented implementation strategies, regulatory ambiguity and varied technical readiness across clinical research sites. To address these systemic challenges, the European Institute for Innovation through Health Data (i~HD) launched the eSource Scale-Up Task Force in 2024.

This cross-industry initiative brings together key stakeholders – academic research centres, hospitals, sponsors and regulatory experts – to drive a unified, scalable approach to eSource implementation, starting with oncology and expanding to other therapeutic areas. i~HD's experience in data interoperability, multi-stakeholder

engagements and governance through initiatives like EHR4CR, EHR2EDC and EU-PEARL (Dupont et al., 2018; Ammo et al., 2023; Lombardo et al., 2023) laid a strong foundation for the Task Force. Its neutral, vendor-agnostic stance ensures collaborative engagement and practical, evidence-driven progress. Membership is invitation-based and designed to foster open dialogue while maintaining confidentiality where required.

While the framework was co-developed with oncology sites and sponsors, its structure – anchored in readiness criteria, KPIs and phased playbook guidance – was intentionally designed to be domain-agnostic. Transfer to other therapeutic areas does not require a new model, but rather adaptation of disease-specific data elements, workflows and regulatory considerations. For example, oncology-centric metrics such as imaging or biomarker data flows can be replaced with cardiology-specific endpoints or neurology-focused assessments while preserving the same governance, interoperability and implementation structures.

The Task Force's core members include Cambridge University Hospitals, Mayo Clinic, Memorial Sloan Kettering Cancer Center and University Hospital of Essen, alongside leading pharmaceutical companies – AstraZeneca, Johnson & Johnson, Lilly, Regeneron and Sanofi (Figure 2). These organizations bring operational and strategic expertise across clinical operations, informatics and data science. The selection of these sponsors and hospitals as core team members is based on their demonstrated leadership: they are either actively conducting eSource-enabled trials or are in the advanced stages of implementing EHR-to-EDC integration. Their hands-on experience ensures that the Task Force's tools and recommendations are grounded in real-world operational contexts.

The Task Force focuses on several high-impact domains:

- **Site Enablement**: Building institutional capacity through readiness assessments, workflow alignment and staff training.
- **Data Interoperability**: Standardizing clinical data exchange using HL7® FHIR® and guidance on common data elements (CDE) to facilitate cross-platform integration.
- **Regulatory Engagement**: Aligning efforts with global authorities (e.g., FDA, EMA) to interpret eSource guidance under Good Clinical Practices (GCP), GDPR, HIPAA and International Council for Harmonization of Technical Requirements for Pharmaceuticals for Human Use (ICH) frameworks.
- **Implementation Science**: Capturing lessons learned from early adopters and embedding them into repeatable, context-sensitive models.

A distinctive feature of the Task Force is the creation of dedicated reference groups – one for sites and one for industry sponsors. Open to organizations progressing toward eSourcing, these groups function as communities of practice where members validate tools, troubleshoot barriers and co-develop scale-up strategies. As membership grows in 2025, they will play a pivotal role in guiding broader adoption. In parallel, the Technical Vendor Reference Group (launching Q3 2025) will convene EHR vendors, EDC providers, middleware developers and system integrators to ensure digital infrastructures align with Task Force deliverables and scale effectively across diverse ecosystems.

The Task Force is also developing modular deliverables, including operational frameworks and a set of Playbook annexes. These annexes serve as practical supplements to the core Playbook, offering detailed templates, checklists and guidance documents to help sites and sponsors address common barriers in implementation. Examples include contracting workflows, ethics and IRB review alignment, value case articulation and strategies for integrating AI

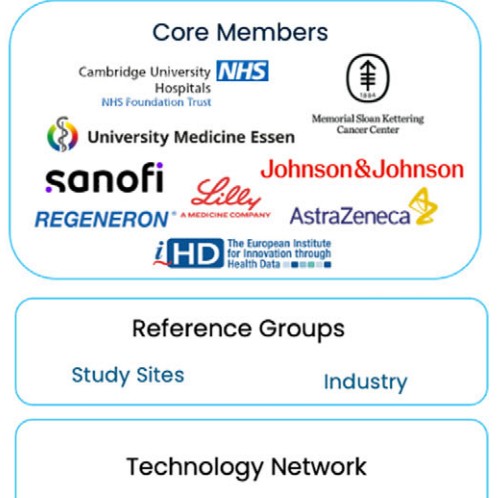

**Figure 2.** The purpose and core members of the i~HD Scale Up Task Force.

tools to manage unstructured data. Together, these resources are designed to ensure that eSource adoption is scalable, sustainable and compliant across diverse trial ecosystems.

Governance is rooted in transparency and neutrality. i~HD leads the initiative with a clear mandate to build consensus while ensuring data protection and regulatory integrity. Its structure enables sponsors and sites to align on shared implementation models without privileging any specific technology or commercial entity.

In summary, the i~HD eSource Scale-Up Task Force is forging a collaborative, standards-aligned pathway to modernize clinical trial execution. By engaging early adopters, enabling cross-sector learning and producing reusable implementation tools, the initiative lays a strong foundation for a digitally integrated research infrastructure – transforming not just oncology trials, but the future of clinical research at large.

### Delivering impact: tools, key performance indicators (KPIs) and the eSource playbook

To accelerate the scale-up of eSource in oncology trials, the i~HD eSource Scale-Up Task Force has produced three foundational deliverables that form the basis of a structured roadmap for implementation. These tools address both strategic alignment and operational execution, allowing stakeholders to transition from pilot projects to scalable, repeatable adoption across diverse settings (Table 1).

The first white paper, **Minimum Success Criteria for Early Adopters**, establishes baseline readiness conditions for institutions and sponsors considering eSource deployment. Developed through cross-sector workshops and validated by early adopters, the guide outlines key criteria across four domains: organizational structure, technical capabilities, regulatory compliance and operational capacity. The checklist-style tool has helped sites and sponsors assess their maturity level, identify gaps and initiate focused enablement planning.

The second white paper, **Selected KPIs for eSource Trials**, offers a metrics-driven framework for evaluating the effectiveness of eSource implementation. It defines eight core KPIs – including

**Table 1.** A summary table of the white papers to visually reinforce the deliverables and their scope (Reference to White Papers: https://www.i-hd.eu/our-programmes/esource-for-scaling-up-clinical-trials-programme/publications)

| White paper | Purpose | Key features |
|---|---|---|
| Minimum Success Criteria *(Jul 2024)* | Define baseline conditions for initiating eSource trials | Readiness checklist across organizational, technical, compliance and staffing domains |
| Selected KPIs *(Dec 2024)* | Measure success of eSource implementations | 8 standardized indicators including data accuracy, SDV reduction, CRC burden and mapping reusability |
| eSource Playbook *(Mar 2025)* | Provide a phased, step-by-step implementation guide for sponsors and sites | 5-phase model (Preparation, Planning, Setup, Execution, Review); tailored workflows and CDE guidance |

accuracy of data transfer, completeness of data mapping, SDV reduction, site efficiency, CRC satisfaction and mapping reusability – that provide sponsors and sites with objective measures to benchmark impact and guide continuous improvement.

The third and most comprehensive deliverable, **The eSource Playbook**, provides a step-by-step operational guide to implementing EHR-to-EDC integration at both sponsor and site levels. Organized into five key phases – preparation, planning, setup, execution and post-implementation review – the Playbook includes decision support tools, role-based workflows and recommendations for managing structured and unstructured data. The Playbook also introduces a shared catalogue of Common Data Elements (CDE) and highlights interoperability standards aligned with HL7® FHIR®.

Importantly, each of these white papers has been reviewed and validated by core Task Force members, including hospitals and sponsors actively engaged in eSource implementation. Early adopter sites have already used the readiness criteria to benchmark institutional capacity, piloted the KPI framework to monitor

## Key areas for collaboration

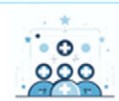

### Site Readiness

- Direct Data transfer readiness requirements for hospitals
- Readiness checklist
- Awareness/education webinars to sites

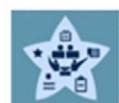

### Industry Playbook

- Establish a playbook for deploying EHR direct data transfer to Sponsor
- Define common data elements catalogue
- Define minimum requirements for EHR to Sponsor solutions

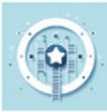

### Regulatory

- Clarifying regulatory guidance interpretations,
- Exploring acceptable approaches and validation rules to collect regulatory-grade unstructured data using eSource technology

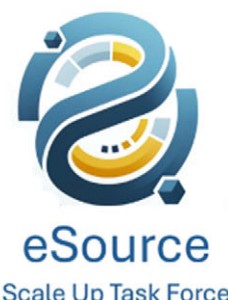

**eSource**
Scale Up Task Force

**Figure 3.** The 2025 Scope of the i~HD Scale Up Task Force.

efficiency and data quality and provided feedback that shaped the Playbook's phased guidance. This collaborative validation ensures that the tools are grounded in real-world operational experience rather than theory alone.

In addition to these core resources, the Playbook is supported by a growing set of annexes, developed to address deeper implementation needs. These include templates for data flow architecture, contracting and vendor management, regulatory validation protocols, Institutional Review Board (IRB)/ethics communication toolkits and AI integration strategies. As such, the Playbook functions as a living document, continuously refined by lessons from ongoing trials and expanded through member contributions.

Each white paper is publicly available through the i~HD platform and designed for modular adoption. Together, these white papers provide a structured, scalable roadmap for eSource adoption. They support cross-functional alignment, foster regulatory confidence and offer practical tools that enable sponsors and sites to move from aspiration to execution. By following this phased approach, institutions can scale eSource adoption confidently and consistently – advancing a new standard in oncology research and beyond.

## Conclusion: scaling for the future of clinical research

The growing complexity and cost of oncology clinical trials have underscored the limitations of manual data workflows. Manual EHR-to-EDC transcription, once standard practice, now delays timelines, increases site burden and compromises data quality. eSource technology offers a scalable, modern alternative – automating data flow, improving accuracy and enhancing operational efficiency.

To overcome barriers to adoption, the i~HD eSource Scale-Up Task Force convened a multi-stakeholder network of hospitals and sponsors, including their clinical operations, informatics and digital health experts. The Task Force developed a vendor-neutral implementation framework anchored by three foundational tools: readiness criteria, performance KPIs and a phased operational Playbook. These resources support the transition from fragmented pilot projects to sustainable, system-wide implementation (Figure 3).

Although designed for oncology, the framework is adaptable to other high-data therapeutic areas, including neurology, cardiology and rare diseases. Expanded Playbook annexes address

interoperability, AI-supported data extraction and ethics review, aligning with evolving global regulatory expectations. As adoption grows, this initiative supports a shift toward digitally enabled, patient-centred clinical trials.

The question is no longer whether eSource will transform clinical trials – but how quickly. If scaled effectively, eSource technologies will not only optimize today's research – but also enable studies that might otherwise never happen, accelerating access to therapies for patients who need them most.

**Open peer review.** To view the open peer review materials for this article, please visit http://doi.org/10.1017/pcm.2025.10004.

**Data availability statement.** This is a perspective article that synthesizes previously published work and implementation experience; no datasets were created or analyzed. All sources cited are publicly available in the References.

**Acknowledgements.** The authors thank the European Institute for Innovation through Health Data (i~HD) for facilitating the eSource Task Force meetings and coordination activities. The authors also acknowledge the contributions of participating hospitals and sponsor organizations who shared their experiences and insights to support the development of the frameworks presented in this article.

**Author contribution.** Conceptualization: Mats Sundgren, Christophe Maes, Dipak Kalra; Methodology: Tracy Acito, Angela Fritsche, Joseph Lengfellner, Sarah Burge; Validation: Amy Cramer, Lars Fransson, Felix Nensa, Pavithra Mariappan; Writing – Original Draft: Mats Sundgren, Christophe Maes, Sarah Burge, Peter Casteleyn; Writing – Review & Editing: Chris Harrison, Ross Caldow, Peter Casteleyn, Michael Ward; Regulatory and Interoperability Input: Nadir Ammour, Camille Couvert, Michael Ward; Project Administration: Veronique Berthou, Joeri Holtzem, Peter Casteleyn; Clinical Input: Laurie Jackson, Anna Patruno, Isabel Virchow; Implementation Strategy and Coordination: Peter Casteleyn, Fakhry Kaoukdji, Pascal Coorevits; Operational Tools and Playbook Review: Nancy Wetzel, Sharon Klein, Christopher Thompson, Dawn Snow, Robert Green; Review Support: Paul Jacobs; Supervision: Mats Sundgren, Dipak Kalra. All authors reviewed and approved the final manuscript for submission.

**Financial support.** The authors received no external funding for this work. Institutional salaries and routine departmental resources supported authors' time.

**Competing interests.** The authors declare no conflicts of interest relevant to the content of this manuscript. All contributions reflect the views and experiences of the authors in their roles as clinical investigators, sponsor representatives, or healthcare professionals involved in implementing eSource solutions.

None of the authors are affiliated with, or represent, EHR2EDC vendors or data integration technology providers.

**Ethical standard.** This article does not report on original research involving human participants or animals. As such, ethical approval and informed consent were not required.

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
