## [Reviewer Report]

This manuscript addresses a critical bottleneck in modern clinical research—manual and fragmented data workflows in oncology trials—and proposes a pragmatic, collaborative, and scalable solution through the i-HD eSource Scale-Up Task Force initiative. The paper is grounded in real-world experience from leading global institutions. It offers concrete, ready-to-use tools (readiness criteria, KPIs, and an operational Playbook), making it both timely and actionable. Its focus on interoperability, regulatory alignment, and cross-sector engagement positions it as a high-impact contribution to the field. Given the urgency to modernize trial infrastructures amidst rising data complexity, this Perspective offers a visionary yet realistic roadmap that can influence research practices well beyond oncology.

---

## [Reviewer Report]

The manuscript describes the formation of an eSource Task Force with the goal of developing a framework for how clinical trial sites can transition from the traditional, manual approaches currently used for data acquisition and management to a more automated or electronic approach (eSource), specifically direct EHR-to-EDC interoperability, to support automated data extraction. There is no doubt that there is a need for such a framework, and, if effective and scalable, of such a framework’s potential to positively impact and optimize clinical research operations across trial sites. The work described would likely be of great interest to readers. However, there are several items that need to be addressed prior to publication. In general, the focus of the paper should be made clear (either to describe the need for and formation of the Task Force and its objectives and/or to describe the development and validation of the eSource Framework). It is also recommended that the Task Force’s next steps (anticipated milestones and/or additional deliverables and any known timelines) be included and described in more detail (can be at the end of the manuscript) so that the readers know what to expect now that the eSource Framework has been developed.

<b>Grammar & Punctuations:</b> There are a few minor grammar and punctuation items that can be corrected (e.g., missing or unnecessary commas and use of past, present, or future tenses). For example, take a look at the “A Strategic Response: The i~HD eSource Scale-Up Task Force” section and review the use of tenses throughout. Some things that have happened (the formation of the group, decisions already made, etc.) should be stated using past tense.

Page 3, Line 19: In the 2nd paragraph of the “Introduction” – the last sentence: commas can be added around the term “increasingly” so that it reads “…eSource also enables structured and, increasingly, AI-assisted extraction of…”

Page 4, Lines 44–45: In the 1st paragraph of the “Impact on Timelines, Cost, and Trial Viability” – the 3rd sentence: There are 2 items (1) the comma after “sponsors” can be removed so that it reads “For sponsors in competitive oncology indications, such delays...” AND (2) the comma after “market opportunities” can be removed and replaced by “and” so that it reads “…Remove the comma here and replace with an “and”: “...missed market opportunities and reduced investor confidence.”

Page 5, Lines 38–39: In the 2nd paragraph of “The Opportunity: eSource and EHR-to-EDC Integration” – the first sentence: update to add commas “...standards, such as HL7 FHIR and SMART on FHIR APIs, to securely...”

Page 6, Lines 41–42: In the 1st paragraph of “A Strategic Response: The i~HD eSource Scale-Up Task Force” – the sentence that mentions the 3 initiatives: remove the word “such as” so that it reads: “…through initiatives like EHR4CR, EHR2EDC and EU-PEARL…”

Page 7, Line 44: In the paragraph above Figure 3 of “A Strategic Response: The i~HD eSource Scale-Up Task Force” – add a comma after “challenges” so that it reads “These resources will address challenges, such as contracting…”

<b>Figures & Tables:</b>

For Figure 1 – It is recommended that a “note” or “legend” be added as part of this figure to define abbreviations used in the image.

For Figure 1 – It is recommended that the term “Clinical Database” at the end of each method (Existing Methods and Data Automation Technology flows) be updated to something that more clearly indicates this is the study database for research and not a database used for clinical care.

For Figure 1 – It is recommended that the “Data Automation Technology” flow be updated to also start with the “Study Visit” as the first step in the flow, which then moves to the “EHR Data Entry” activity, since eSource does not change that process.

For Figure 2 – A minor correction to the 1st bullet of the Purpose to update to: “Drive and scale the adoption of…”

For Figure 2 – It is recommended that a “note” or “legend” be added as part of this figure to define/describe the purpose of the “Core Members” vs. the “Reference Groups” vs. the “Technology Networks” and/or provide a more detailed description of these roles in the main body of the paper.

<b>Citations & References:</b>

In the References list, there are 2 references that were not used/cited in the paper and should be removed: (1) Lengfellner & Yeatman, 2025 (Page 10, Lines 46–47); and (2) Passut, 2021 (Page 11, Lines 3–5).

Page 11, Line 19: In the References list, the last reference listed as “University of California” should be updated to include the author (Julia Busiek), and the corresponding citation used in the “Impact on Timelines, Cost, and Trial Variability” section (Page 4, Line 53) should be corrected accordingly (to from “University of California, 2025” to “Busiek, 2025”).

Page 4, Line 53: In the “Impact on Timelines, Cost, and Trial Variability” section, the UAB citation needs to be corrected: from “UAB Institute for Human Rights Blog, 2025” to “Rhodes, 2025”.

<b>General Comments:</b>

Page 2, Lines 38–40: In the last paragraph of the Abstract, it is mentioned that “This paper highlights how collaborative governance, phased implementation, and shared operational standards can enable sustainable scaleup of eSource technologies and foster digitally integrated clinical research infrastructures.” This implies that the manuscript will describe not only the formation, purpose, and deliverables of the Task Force and the resulting Framework, but that it will also demonstrate how it can be successfully implemented and scaled up beyond the initial use case. However, this is not demonstrated in this particular article. There is no mention of the actual implementation of the framework – only brief mention that it was an objective and that it was likely developed. There is also no mention of the evaluation of its success nor of any attempts to apply this to other TAs outside of oncology or other to other sites outside of the Core Members. It is suggested that this sentence be removed or modified to more clearly articulate the focus of this paper.

Page 2, Lines 44–52: In the Impact Statement, the authors start by stating that the paper will “present a scalable framework for implementing eSource technologies…” However, while the article mentions a framework was a deliverable and that white papers were published, the actual framework itself is not clearly described. (Unless Figure 3 – and maybe in combination with the “Key Features” in Table 1 – is/are meant to represent the framework – in which case, this should be made clearer.) If the primary objective is to present the final framework, then this needs to be more clearly stated throughout and a section describing the framework in more detail should be included. If the primary objective is to focus more on the Task Force – the need for such a group, its formation, its purpose, goals, and objectives, etc. – then the Impact Statement should be updated to indicate that.

The article starts off by talking through some of the complexities of running oncology trials and mentions that the i~HD Task Force and eSource Framework were initially focused on and/or specific to addressing the challenges faced in the oncology domain. It is then noted that the framework could be applied to other domains. However, in describing the development of the framework, it is not clear if that “oncology” focus shifted once the Task Force started putting all the pieces together (so the resulting framework did end up being more “generic” or broad), or if the framework was created with the oncology focus and “tips” or “amendments” or similar have been (or will be) created for how to “translate” the framework to other domains/therapeutic areas (TAs). Additionally, the statement was made that “While grounded in oncology, the model is transferable to other high-data-density therapeutic areas.” However, there is no real explanation as to how to go about transferring it to other TAs, or even on what elements would need to be modified or reevaluated or considered when translating to other TAs.

Page 2, Line 60: It is suggested that the “Introduction” header be updated to simply “Introduction” and the “Rising Complexity in Oncology Trials” piece be removed - especially since this heading text is repeated as a header for the 2nd paragraph in “The Challenge” section that follows.

Page 3, Lines 5–8: In the 1st paragraph of the “Introduction” – the 2nd sentence: “Traditional trial workflows require research teams to extract, transcribe, and validate patient data from EHRs into EDC systems—a duplicative process that accounts for over 50% of clinical trial data and requires significant verification effort, often consuming substantial operational resources (Kalankesh and Monaghesh, 2024).” – There are 2 questions: (1) the part that reads “…a duplicative process that accounts for over 50% of clinical trial data…” does not make sense. Do you mean that the “duplicative process” accounts for a 50% increase in time required to complete tasks? Or that over 50% of the data in oncology (and other data-intensive domains) are data that are part of a more complicated workflow (manual extraction, transcription, and validation) with redundancies? AND (2) Is this the correct citation? This article by Kalankesh and Monaghesh is a systematic literature review about EHR use in clinical trials. There is nothing in this article that discusses the analysis of any quantitative data that would account for this “50%” value presented here. One could also argue that this particular systematic literature review is relatively weak – as the search criteria is very broad, the numbers presented in the results do not match the figure displayed (nor does it seem remotely accurate based on the very broad search criteria), and there is practically no synthesis of the identified articles – which is the main point of conducting and writing up a review. It is recommended that this citation be replaced with one that contains this information and can back up the “50%” value in some way; or to modify the sentence so that a specific percentage is not included (and remove the citation completely).

Page 3, Lines 17–21: In the 2nd paragraph of the “Introduction” – the last sentence: The 3 citations come at the end of the sentence, and by citing these articles here, it implies that they discuss standards-based (or FHIR-based) eSource data extraction. They do focus on AI for unstructured data, but all 3 do not necessarily focus on AI-based eSource or AI to support eSource extraction, so it does not seem like they fit here. It may be more appropriate to move them to come after the text: “...increasingly, AI-assisted (citations) extraction of clinical data...”

Page 3, Lines 52–54: In the 1st paragraph of “The Challenge” – the last sentence: “This duplication affects more than half of the total trial data and contributes to roughly 20% of total study costs—resources that could be redirected toward scientific advancement and patient benefit (Kalankesh and Monaghesh, 2024).” – Again, this citation does not seem to be appropriate to back up the data presented in this sentence. If an article was identified by this systematic review that contains this “20%” value, that article should be cited instead of this review. Otherwise, it is recommended that this sentence be removed or rewritten to exclude the percentage and the citation should be deleted (and removed from the reference list, if it is no longer used in the article).

Page 4, Lines 3–6: In the 1st paragraph of “Increasing Data Complexity in Modern Oncology Trials” – the last 2 sentences: “In some advanced trials, the number of data points collected per patient has increased exponentially—from 10,000 in traditional Phase III studies to over 100,000 in trials involving genomic and digital health data. Each of these data points must be documented, reviewed, and often verified manually (Altomare et al., 2024).” – There are 2 items: (1) It would seem more likely or appropriate that a citation would be needed to back up the “10,000 to over 100,000 data points” claim, as opposed to being used for the last sentence in this paragraph. AND (2) That said, Is this Altamore et al. citation the correct or appropriate citation? (Is this citation even needed at all?) In reviewing the Altomare et al. abstract (unable to find a full text version), it does not appear to match what is written in these 2 sentences. It is recommended that the Altomare et al. citation be removed and the reference deleted, as this is the only time it is used in this manuscript.

Page 4, Line 19: In the 1st paragraph of “The Burden of Redundant Data Entry” – the last sentence: Is this meant to point to the Sundgren et al., 2025 citation (rather than the current 2024 citation), as the 2025 article mentions the “high-value tasks” (or maybe cite both if both are relevant).

Page 4, Line 28: In the 2nd paragraph of “The Burden of Redundant Data Entry” – the last sentence: “In addition to transcription itself, manual data entry generates substantial downstream workload, including query resolution, data reconciliation, and extensive data review cycles—activities that are estimated to account for up to 30–40% of total data management costs in oncology trials (Hamidi et al., 2024; Ehidiamen and Oladapo, 2024). – There are 2 items: (1) The Hamidi et al. article mentions SDV-related costs (TA-agnostic) as being attributable to between 25%-40% of clinical trial costs. It is recommended that the 30-40% mentioned in the sentence be updated to align with the citated article. AND (2) The Ehidiamen & Oladapo article does not discuss data management costs in oncology trials. It is a non-therapeutic area specific (TA-agnostic) paper on EDC use for/in clinical trials. If this is meant to refer to the “substantial downstream workload”, it is recommended that this be moved to follow the text “...extensive data review cycles (E & O citation)--activities that are...” Otherwise, it is recommended that this citation be removed.

Page 4, Lines 31–35: In the “Compliance and Data Integrity Challenges” paragraph – the 2nd sentence: “Standards such as ALCOA+ (Attributable, Legible, Contemporaneous, Original, Accurate, and Complete) and compliance frameworks like the Global Data Protection Regulation (GDPR) and the Health Insurance Portability and Accountability Act (HIPAA) underscore the necessity of data that is not only accurate but secured and auditable (Ehidiamen and Oladapo, 2024).” – Is this the correct or appropriate citation for this sentence? There is mention of GDPR and HIPAA in this article, but it points out to other references re: those requirements. There is no mention of ALCOA+. Since this article is not the direct source for info on these regulations, it is not recommended that it be used as a citation for this sentence. Better to cite the actual regulations or an article (or two) that specifically discuss these regulations. If the E & O citation is removed here and removed from the previous section (based on the previous comment), it is recommended that it also be removed from the Reference list, as those are the only 2 places it is cited.

Page 4, Lines 45–46: In the “Impact on Timelines, Cost, and Trial Viability” paragraph – the 4th sentence: “For patients, lengthy timelines increase the risk for patient safety identification as well as limited access to breakthrough therapies.” – This sentence does not read clearly. Perhaps update to: “...increase the risk for patient safety identification and limit access to breakthrough therapies.” OR “...increase the risk for patient safety identification, as well as limit access to breakthrough therapies.”

Page 5, Lines 45–51: In the 3rd paragraph of “The Opportunity: eSource and EHR-to-EDC Integration” – There are 3 items: (1) In the 2nd sentence re: automation resulting in “higher data quality and completeness” – If available, it may be worth adding a citation or two to support this statement to further emphasize that this has been shown/demonstrated and is not just something we “hope” or “assume” will happen. AND (2) In the 3rd sentence re: FDA, EMA, and MHRA support for eSource – Can you point to guidances or press releases or something similar from these agencies that express their increased support? AND (3) In the last sentence re: “direct cost and time savings” – If available, it may be worth adding a citation or two to support this statement, again, to show or demonstrate that sites have experienced these improvements.

Page 5, Lines 53–54: In the 4th paragraph of “The Opportunity: eSource and EHR-to-EDC Integration” – the 1st sentence: Consider not limiting the statement to only RCTs. The “eSource transfer process” is applicable to all research requiring extensive amounts of data from outside sources (not just RCTs).

Page 6, Lines 3–8: In the 5th paragraph of “The Opportunity: eSource and EHR-to-EDC Integration” – There are 3 items: (1) In the 2nd sentence re: Mayo Clinic – This might benefit from a citation, if available. AND (2) In the 3rd sentence re: City of Hope – This might also benefit from a citation, if available. AND (3) In the last sentence –Unless these institutions utilized this particular “structured and collaborative framework” (it is not clear by the term “early adopters” if that means of the framework or of eSource in general) and (ideally) have the data to back up their improvements, this statement about “scalability and reproducibility” cannot be made. Also, one could argue that you would need more than 2 examples and a wider variety of site types for this to truly be the case.

Page 6, Lines 15–21: In the 7th paragraph of “The Opportunity: eSource and EHR-to-EDC Integration” – the 2nd sentence that mentions EHR2EDC, TransCelerate, and FDA RWE – There are 3 items: (1) If the citations included are meant to be citations for the EHR2EDC (Mueller et al, and Ammour et al), TransCelerate eSource Project, and FDA RWE, it is recommended that they be moved to come right after the corresponding project. AND (2) Also – the Claerhout et al. article seems to be about EHR4CR, which is not mentioned here, but is mentioned in the 1st paragraph of “A Strategic Response” section. (Was not sure if this should be moved there.) AND (3) Are there citations available for the TransCelerate eSource project (even if just a webpage)? Could also point to the FDA’s RWE guidance as a citation here.

Page 6, Line 23: In the last paragraph of “The Opportunity: eSource and EHR-to-EDC Integration” there is mention of “a scalable solution” – Refer to comment above re: scalability. It is recommended that the term “scalable” be removed here and leaving it as: “...offer a future-ready solution to...” The scalability piece is something you could claim about the implementation framework. But, as it stands, and based on what has been presented in this article so far, there is not sufficient evidence to support the scalability claim.

Page 6, Lines 33–34: In the 1st paragraph of “A Strategic Response: The i~HD eSource Scale-Up Task Force” – the 1st sentence: Not sure if this is the best phrase to use because, without sufficient evidence presented to back this up (reduced burden, streamlining of clinical trials), one could argue that it is not so clear. **Not arguing that there is not promise, only emphasizing the need for evidence for readers to see that this is not a statement or assumption, but a truth.

Page 6, Line 46: When describing the Task Force, the term “core members” is used. Are there “non-core” members as well? You mention that the Task Force brought together a variety of key stakeholders, including regulatory experts (and Regulatory Engagement is further emphasized in the description of the “high-impact domains of focus”), but there are no regulators listed in this “core” member group. And what about technology companies (e.g., EHR and EDC vendors)? One could assume they were excluded as “core members” to keep things vendor neutral, but do they get included later as part of the “Technology Networks” mentioned in Figure 2?

Page 7: When elaborating on the content displayed in Figure 2 (and about the Task Force formation in general) – It may be worth adding details on how these groups (Core Members vs. Reference Groups vs. Technology Networks) were identified and selected to be part of the Task Force. Was there a “call to action” or application process? Did i~HD leadership select these members randomly or based on XYZ criteria? More details are needed on the reference groups. How sites/sponsors/members of these groups get selected? Do they meet regularly? What are their required tasks/objectives and/or how do they get engaged/incorporated into the Task Force activities?

Page 7, Lines 25–36: In the 3rd paragraph of “A Strategic Response: The i~HD eSource Scale-Up Task Force” (right after Figure 2) – The bullet re: Implementation Science mentions “early adopters” – this brings to mind the question: Is/was the goal to only implement at high-impact sites? or large AMCs? The core members include heavy hitters that typically have the resources and personnel on board (champions). While they can help with major components/aspects of eSource implementation (and are very necessary stakeholders to include), they are not necessarily indicative of what many other sites might need or encounter on their implementation journeys. How is/was this accounted for?

Page 7, Lines 43–47: In the 5th paragraph of “A Strategic Response: The i~HD eSource Scale-Up Task Force” (right before Figure 3) – More details are needed on the “modular deliverables,” particularly on what is meant by “playbook annexes” to clearly articulate to the reader what they are and what they are/will be used for.

Page 8, Lines 20–29: In the last 2 paragraphs of “A Strategic Response: The i~HD eSource Scale-Up Task Force” (right after Figure 3) – both are a little confusing. Re: Governance paragraph – Governance goes beyond technology use/selection. Also need more details on how the structure enables the shared implementation models. Re: the Summary paragraph – Not sure if this paragraph belongs here (in this section). So far, after reading this section, there are still many questions about the Task Force – its formation, its operations, its anticipated deliverables/milestones…etc. So it is hard to have a summary statement here about how it is forging a pathway to modernize clinical trial execution. It may be better suited for later in the manuscript, after all the details about the Task Force and its activities have been presented.

Page 8, Line 58 and Page 9, Lines 7 and 13: Re: the 3 White Papers – It would be helpful to the reader if a citation for each was noted after each paper is mentioned (with a corresponding reference in the Reference list). Or, the citation could be included in Table 1 within the White Paper column.

Pages 8 and 9: Also re: the 3 White Papers – Were these documents used and “tested” or “validated” by any of the core members (hospitals) or other sites? This is not made clear in this manuscript.

Page 9, Line 30: In the last paragraph of the “Delivering Impact” section – there is a citation (Adamson et al, 2023) that does not seem to fit. This citation does not seem to match what is being stated in the preceding sentence. It has nothing to do with the white papers listed. It is recommended that the citation be removed and the reference be deleted from the Reference List, as this is the only time it is used in the paper.

Page 9, Line 44: In the 2nd paragraph of the “Conclusion” – There is mention of “digital health experts”. Based on the “core members” mentioned, it does not seem like these experts are represented. Are these “digital health experts” already part of or members of the hospitals and/or sponsors, or are they meant to be a different entity completely?

---

## [Reviewer Report]

1. The authors conducted a study to establish a consensus-driven roadmap for eSource adoption. Key deliverables were: readiness criteria for early adopters, a performance indicator framework for monitoring success, and an operational playbook to guide implementation. The eSource Scale-Up Task Force has conducted important work. However, the presentation of that work in this manuscript needs to be improved.

2. The initial 3 ½ text pages appear to be an elongated introduction. This section needs to be reduced to 1 ½ pages at most.

3. The introduction section should address four issues:

1. Background of the research question

2. Previous research in the area

3. Problems with past research

4. What you did to fix those problems

4. Critical sections in the introductory material are presented without appropriate references. Are these the opinions of Task Force members? If so, they should be identified as such.

5. The presentation of methods and results covers only 3 pages, including figures. This section needs more detail. Readers will want to know how the Task Force approached their work and what were the important findings.

6. This study relies upon the expertise of Task Force members. Merely citing organizational affiliations is not sufficient to establish their expertise for readers. More information on their relevant eSource and other experiences is needed. For example, how many eSource studies have they conducted?

7. How did the Task Force develop their three deliverables? Were there teams? Did the teams meet? If so, how many meetings? Who led this work? More details are needed to assure readers of the correctness of the methods used.

8. This study’s three deliverables should be referenced with URLs. Nonetheless, the authors should describe the content of these deliverables in depth in their manuscript.

---

## [Editor Report]

Dear Authors, 

the decision has been made to accept the paper, however the reivewers would like to see full attribution and COI statements for the authors, and several changes and alterations (of a grammar, clarity and other nature below)

These changes should be relatively straightforward to make.

The manuscript describes the formation of an eSource Task Force with the goal of developing a framework for how clinical trial sites can transition from the traditional, manual approaches currently used for data acquisition and management to a more automated or electronic approach (eSource), specifically direct EHR-to-EDC interoperability, to support automated data extraction. There is no doubt that there is a need for such a framework, and, if effective and scalable, of such a framework’s potential to positively impact and optimize clinical research operations across trial sites. The work described would likely be of great interest to readers. However, there are several items that need to be addressed prior to publication. In general, the focus of the paper should be made clear (either to describe the need for and formation of the Task Force and its objectives and/or to describe the development and validation of the eSource Framework). It is also recommended that the Task Force’s next steps (anticipated milestones and/or additional deliverables and any known timelines) be included and described in more detail (can be at the end of the manuscript) so that the readers know what to expect now that the eSource Framework has been developed.



Grammar & Punctuations: There are a few minor grammar and punctuation items that can be corrected (e.g., missing or unnecessary commas and use of past, present, or future tenses). For example, take a look at the “A Strategic Response: The i~HD eSource Scale-Up Task Force” section and review the use of tenses throughout. Some things that have happened (the formation of the group, decisions already made, etc.) should be stated using past tense.



Page 3, Line 19: In the 2nd paragraph of the “Introduction” – the last sentence: commas can be added around the term “increasingly” so that it reads “…eSource also enables structured and, increasingly, AI-assisted extraction of…”



Page 4, Lines 44–45: In the 1st paragraph of the “Impact on Timelines, Cost, and Trial Viability” – the 3rd sentence: There are 2 items (1) the comma after “sponsors” can be removed so that it reads “For sponsors in competitive oncology indications, such delays...” AND (2) the comma after “market opportunities” can be removed and replaced by “and” so that it reads “…Remove the comma here and replace with an “and”: “...missed market opportunities and reduced investor confidence.”



Page 5, Lines 38–39: In the 2nd paragraph of “The Opportunity: eSource and EHR-to-EDC Integration” – the first sentence: update to add commas “...standards, such as HL7 FHIR and SMART on FHIR APIs, to securely...”



Page 6, Lines 41–42: In the 1st paragraph of “A Strategic Response: The i~HD eSource Scale-Up Task Force” – the sentence that mentions the 3 initiatives: remove the word “such as” so that it reads: “…through initiatives like EHR4CR, EHR2EDC and EU-PEARL…”



Page 7, Line 44: In the paragraph above Figure 3 of “A Strategic Response: The i~HD eSource Scale-Up Task Force” – add a comma after “challenges” so that it reads “These resources will address challenges, such as contracting…”



Figures & Tables:



For Figure 1 – It is recommended that a “note” or “legend” be added as part of this figure to define abbreviations used in the image.



For Figure 1 – It is recommended that the term “Clinical Database” at the end of each method (Existing Methods and Data Automation Technology flows) be updated to something that more clearly indicates this is the study database for research and not a database used for clinical care.



For Figure 1 – It is recommended that the “Data Automation Technology” flow be updated to also start with the “Study Visit” as the first step in the flow, which then moves to the “EHR Data Entry” activity, since eSource does not change that process.



For Figure 2 – A minor correction to the 1st bullet of the Purpose to update to: “Drive and scale the adoption of…”



For Figure 2 – It is recommended that a “note” or “legend” be added as part of this figure to define/describe the purpose of the “Core Members” vs. the “Reference Groups” vs. the “Technology Networks” and/or provide a more detailed description of these roles in the main body of the paper.



Citations & References:



In the References list, there are 2 references that were not used/cited in the paper and should be removed: (1) Lengfellner & Yeatman, 2025 (Page 10, Lines 46–47); and (2) Passut, 2021 (Page 11, Lines 3–5).



Page 11, Line 19: In the References list, the last reference listed as “University of California” should be updated to include the author (Julia Busiek), and the corresponding citation used in the “Impact on Timelines, Cost, and Trial Variability” section (Page 4, Line 53) should be corrected accordingly (to from “University of California, 2025” to “Busiek, 2025”).



Page 4, Line 53: In the “Impact on Timelines, Cost, and Trial Variability” section, the UAB citation needs to be corrected: from “UAB Institute for Human Rights Blog, 2025” to “Rhodes, 2025”.



General Comments:



Page 2, Lines 38–40: In the last paragraph of the Abstract, it is mentioned that “This paper highlights how collaborative governance, phased implementation, and shared operational standards can enable sustainable scaleup of eSource technologies and foster digitally integrated clinical research infrastructures.” This implies that the manuscript will describe not only the formation, purpose, and deliverables of the Task Force and the resulting Framework, but that it will also demonstrate how it can be successfully implemented and scaled up beyond the initial use case. However, this is not demonstrated in this particular article. There is no mention of the actual implementation of the framework – only brief mention that it was an objective and that it was likely developed. There is also no mention of the evaluation of its success nor of any attempts to apply this to other TAs outside of oncology or other to other sites outside of the Core Members. It is suggested that this sentence be removed or modified to more clearly articulate the focus of this paper.



Page 2, Lines 44–52: In the Impact Statement, the authors start by stating that the paper will “present a scalable framework for implementing eSource technologies…” However, while the article mentions a framework was a deliverable and that white papers were published, the actual framework itself is not clearly described. (Unless Figure 3 – and maybe in combination with the “Key Features” in Table 1 – is/are meant to represent the framework – in which case, this should be made clearer.) If the primary objective is to present the final framework, then this needs to be more clearly stated throughout and a section describing the framework in more detail should be included. If the primary objective is to focus more on the Task Force – the need for such a group, its formation, its purpose, goals, and objectives, etc. – then the Impact Statement should be updated to indicate that.



The article starts off by talking through some of the complexities of running oncology trials and mentions that the i~HD Task Force and eSource Framework were initially focused on and/or specific to addressing the challenges faced in the oncology domain. It is then noted that the framework could be applied to other domains. However, in describing the development of the framework, it is not clear if that “oncology” focus shifted once the Task Force started putting all the pieces together (so the resulting framework did end up being more “generic” or broad), or if the framework was created with the oncology focus and “tips” or “amendments” or similar have been (or will be) created for how to “translate” the framework to other domains/therapeutic areas (TAs). Additionally, the statement was made that “While grounded in oncology, the model is transferable to other high-data-density therapeutic areas.” However, there is no real explanation as to how to go about transferring it to other TAs, or even on what elements would need to be modified or reevaluated or considered when translating to other TAs.



Page 2, Line 60: It is suggested that the “Introduction” header be updated to simply “Introduction” and the “Rising Complexity in Oncology Trials” piece be removed - especially since this heading text is repeated as a header for the 2nd paragraph in “The Challenge” section that follows.



Page 3, Lines 5–8: In the 1st paragraph of the “Introduction” – the 2nd sentence: “Traditional trial workflows require research teams to extract, transcribe, and validate patient data from EHRs into EDC systems—a duplicative process that accounts for over 50% of clinical trial data and requires significant verification effort, often consuming substantial operational resources (Kalankesh and Monaghesh, 2024).” – There are 2 questions: (1) the part that reads “…a duplicative process that accounts for over 50% of clinical trial data…” does not make sense. Do you mean that the “duplicative process” accounts for a 50% increase in time required to complete tasks? Or that over 50% of the data in oncology (and other data-intensive domains) are data that are part of a more complicated workflow (manual extraction, transcription, and validation) with redundancies? AND (2) Is this the correct citation? This article by Kalankesh and Monaghesh is a systematic literature review about EHR use in clinical trials. There is nothing in this article that discusses the analysis of any quantitative data that would account for this “50%” value presented here. One could also argue that this particular systematic literature review is relatively weak – as the search criteria is very broad, the numbers presented in the results do not match the figure displayed (nor does it seem remotely accurate based on the very broad search criteria), and there is practically no synthesis of the identified articles – which is the main point of conducting and writing up a review. It is recommended that this citation be replaced with one that contains this information and can back up the “50%” value in some way; or to modify the sentence so that a specific percentage is not included (and remove the citation completely).



Page 3, Lines 17–21: In the 2nd paragraph of the “Introduction” – the last sentence: The 3 citations come at the end of the sentence, and by citing these articles here, it implies that they discuss standards-based (or FHIR-based) eSource data extraction. They do focus on AI for unstructured data, but all 3 do not necessarily focus on AI-based eSource or AI to support eSource extraction, so it does not seem like they fit here. It may be more appropriate to move them to come after the text: “...increasingly, AI-assisted (citations) extraction of clinical data...”



Page 3, Lines 52–54: In the 1st paragraph of “The Challenge” – the last sentence: “This duplication affects more than half of the total trial data and contributes to roughly 20% of total study costs—resources that could be redirected toward scientific advancement and patient benefit (Kalankesh and Monaghesh, 2024).” – Again, this citation does not seem to be appropriate to back up the data presented in this sentence. If an article was identified by this systematic review that contains this “20%” value, that article should be cited instead of this review. Otherwise, it is recommended that this sentence be removed or rewritten to exclude the percentage and the citation should be deleted (and removed from the reference list, if it is no longer used in the article).



Page 4, Lines 3–6: In the 1st paragraph of “Increasing Data Complexity in Modern Oncology Trials” – the last 2 sentences: “In some advanced trials, the number of data points collected per patient has increased exponentially—from 10,000 in traditional Phase III studies to over 100,000 in trials involving genomic and digital health data. Each of these data points must be documented, reviewed, and often verified manually (Altomare et al., 2024).” – There are 2 items: (1) It would seem more likely or appropriate that a citation would be needed to back up the “10,000 to over 100,000 data points” claim, as opposed to being used for the last sentence in this paragraph. AND (2) That said, Is this Altamore et al. citation the correct or appropriate citation? (Is this citation even needed at all?) In reviewing the Altomare et al. abstract (unable to find a full text version), it does not appear to match what is written in these 2 sentences. It is recommended that the Altomare et al. citation be removed and the reference deleted, as this is the only time it is used in this manuscript.



Page 4, Line 19: In the 1st paragraph of “The Burden of Redundant Data Entry” – the last sentence: Is this meant to point to the Sundgren et al., 2025 citation (rather than the current 2024 citation), as the 2025 article mentions the “high-value tasks” (or maybe cite both if both are relevant).



Page 4, Line 28: In the 2nd paragraph of “The Burden of Redundant Data Entry” – the last sentence: “In addition to transcription itself, manual data entry generates substantial downstream workload, including query resolution, data reconciliation, and extensive data review cycles—activities that are estimated to account for up to 30–40% of total data management costs in oncology trials (Hamidi et al., 2024; Ehidiamen and Oladapo, 2024). – There are 2 items: (1) The Hamidi et al. article mentions SDV-related costs (TA-agnostic) as being attributable to between 25%-40% of clinical trial costs. It is recommended that the 30-40% mentioned in the sentence be updated to align with the citated article. AND (2) The Ehidiamen & Oladapo article does not discuss data management costs in oncology trials. It is a non-therapeutic area specific (TA-agnostic) paper on EDC use for/in clinical trials. If this is meant to refer to the “substantial downstream workload”, it is recommended that this be moved to follow the text “...extensive data review cycles (E & O citation)--activities that are...” Otherwise, it is recommended that this citation be removed.



Page 4, Lines 31–35: In the “Compliance and Data Integrity Challenges” paragraph – the 2nd sentence: “Standards such as ALCOA+ (Attributable, Legible, Contemporaneous, Original, Accurate, and Complete) and compliance frameworks like the Global Data Protection Regulation (GDPR) and the Health Insurance Portability and Accountability Act (HIPAA) underscore the necessity of data that is not only accurate but secured and auditable (Ehidiamen and Oladapo, 2024).” – Is this the correct or appropriate citation for this sentence? There is mention of GDPR and HIPAA in this article, but it points out to other references re: those requirements. There is no mention of ALCOA+. Since this article is not the direct source for info on these regulations, it is not recommended that it be used as a citation for this sentence. Better to cite the actual regulations or an article (or two) that specifically discuss these regulations. If the E & O citation is removed here and removed from the previous section (based on the previous comment), it is recommended that it also be removed from the Reference list, as those are the only 2 places it is cited.



Page 4, Lines 45–46: In the “Impact on Timelines, Cost, and Trial Viability” paragraph – the 4th sentence: “For patients, lengthy timelines increase the risk for patient safety identification as well as limited access to breakthrough therapies.” – This sentence does not read clearly. Perhaps update to: “...increase the risk for patient safety identification and limit access to breakthrough therapies.” OR “...increase the risk for patient safety identification, as well as limit access to breakthrough therapies.”



Page 5, Lines 45–51: In the 3rd paragraph of “The Opportunity: eSource and EHR-to-EDC Integration” – There are 3 items: (1) In the 2nd sentence re: automation resulting in “higher data quality and completeness” – If available, it may be worth adding a citation or two to support this statement to further emphasize that this has been shown/demonstrated and is not just something we “hope” or “assume” will happen. AND (2) In the 3rd sentence re: FDA, EMA, and MHRA support for eSource – Can you point to guidances or press releases or something similar from these agencies that express their increased support? AND (3) In the last sentence re: “direct cost and time savings” – If available, it may be worth adding a citation or two to support this statement, again, to show or demonstrate that sites have experienced these improvements.



Page 5, Lines 53–54: In the 4th paragraph of “The Opportunity: eSource and EHR-to-EDC Integration” – the 1st sentence: Consider not limiting the statement to only RCTs. The “eSource transfer process” is applicable to all research requiring extensive amounts of data from outside sources (not just RCTs).



Page 6, Lines 3–8: In the 5th paragraph of “The Opportunity: eSource and EHR-to-EDC Integration” – There are 3 items: (1) In the 2nd sentence re: Mayo Clinic – This might benefit from a citation, if available. AND (2) In the 3rd sentence re: City of Hope – This might also benefit from a citation, if available. AND (3) In the last sentence –Unless these institutions utilized this particular “structured and collaborative framework” (it is not clear by the term “early adopters” if that means of the framework or of eSource in general) and (ideally) have the data to back up their improvements, this statement about “scalability and reproducibility” cannot be made. Also, one could argue that you would need more than 2 examples and a wider variety of site types for this to truly be the case.



Page 6, Lines 15–21: In the 7th paragraph of “The Opportunity: eSource and EHR-to-EDC Integration” – the 2nd sentence that mentions EHR2EDC, TransCelerate, and FDA RWE – There are 3 items: (1) If the citations included are meant to be citations for the EHR2EDC (Mueller et al, and Ammour et al), TransCelerate eSource Project, and FDA RWE, it is recommended that they be moved to come right after the corresponding project. AND (2) Also – the Claerhout et al. article seems to be about EHR4CR, which is not mentioned here, but is mentioned in the 1st paragraph of “A Strategic Response” section. (Was not sure if this should be moved there.) AND (3) Are there citations available for the TransCelerate eSource project (even if just a webpage)? Could also point to the FDA’s RWE guidance as a citation here.



Page 6, Line 23: In the last paragraph of “The Opportunity: eSource and EHR-to-EDC Integration” there is mention of “a scalable solution” – Refer to comment above re: scalability. It is recommended that the term “scalable” be removed here and leaving it as: “...offer a future-ready solution to...” The scalability piece is something you could claim about the implementation framework. But, as it stands, and based on what has been presented in this article so far, there is not sufficient evidence to support the scalability claim.



Page 6, Lines 33–34: In the 1st paragraph of “A Strategic Response: The i~HD eSource Scale-Up Task Force” – the 1st sentence: Not sure if this is the best phrase to use because, without sufficient evidence presented to back this up (reduced burden, streamlining of clinical trials), one could argue that it is not so clear. **Not arguing that there is not promise, only emphasizing the need for evidence for readers to see that this is not a statement or assumption, but a truth.



Page 6, Line 46: When describing the Task Force, the term “core members” is used. Are there “non-core” members as well? You mention that the Task Force brought together a variety of key stakeholders, including regulatory experts (and Regulatory Engagement is further emphasized in the description of the “high-impact domains of focus”), but there are no regulators listed in this “core” member group. And what about technology companies (e.g., EHR and EDC vendors)? One could assume they were excluded as “core members” to keep things vendor neutral, but do they get included later as part of the “Technology Networks” mentioned in Figure 2?



Page 7: When elaborating on the content displayed in Figure 2 (and about the Task Force formation in general) – It may be worth adding details on how these groups (Core Members vs. Reference Groups vs. Technology Networks) were identified and selected to be part of the Task Force. Was there a “call to action” or application process? Did i~HD leadership select these members randomly or based on XYZ criteria? More details are needed on the reference groups. How sites/sponsors/members of these groups get selected? Do they meet regularly? What are their required tasks/objectives and/or how do they get engaged/incorporated into the Task Force activities?



Page 7, Lines 25–36: In the 3rd paragraph of “A Strategic Response: The i~HD eSource Scale-Up Task Force” (right after Figure 2) – The bullet re: Implementation Science mentions “early adopters” – this brings to mind the question: Is/was the goal to only implement at high-impact sites? or large AMCs? The core members include heavy hitters that typically have the resources and personnel on board (champions). While they can help with major components/aspects of eSource implementation (and are very necessary stakeholders to include), they are not necessarily indicative of what many other sites might need or encounter on their implementation journeys. How is/was this accounted for?



Page 7, Lines 43–47: In the 5th paragraph of “A Strategic Response: The i~HD eSource Scale-Up Task Force” (right before Figure 3) – More details are needed on the “modular deliverables,” particularly on what is meant by “playbook annexes” to clearly articulate to the reader what they are and what they are/will be used for.



Page 8, Lines 20–29: In the last 2 paragraphs of “A Strategic Response: The i~HD eSource Scale-Up Task Force” (right after Figure 3) – both are a little confusing. Re: Governance paragraph – Governance goes beyond technology use/selection. Also need more details on how the structure enables the shared implementation models. Re: the Summary paragraph – Not sure if this paragraph belongs here (in this section). So far, after reading this section, there are still many questions about the Task Force – its formation, its operations, its anticipated deliverables/milestones…etc. So it is hard to have a summary statement here about how it is forging a pathway to modernize clinical trial execution. It may be better suited for later in the manuscript, after all the details about the Task Force and its activities have been presented.



Page 8, Line 58 and Page 9, Lines 7 and 13: Re: the 3 White Papers – It would be helpful to the reader if a citation for each was noted after each paper is mentioned (with a corresponding reference in the Reference list). Or, the citation could be included in Table 1 within the White Paper column.



Pages 8 and 9: Also re: the 3 White Papers – Were these documents used and “tested” or “validated” by any of the core members (hospitals) or other sites? This is not made clear in this manuscript.



Page 9, Line 30: In the last paragraph of the “Delivering Impact” section – there is a citation (Adamson et al, 2023) that does not seem to fit. This citation does not seem to match what is being stated in the preceding sentence. It has nothing to do with the white papers listed. It is recommended that the citation be removed and the reference be deleted from the Reference List, as this is the only time it is used in the paper.



Page 9, Line 44: In the 2nd paragraph of the “Conclusion” – There is mention of “digital health experts”. Based on the “core members” mentioned, it does not seem like these experts are represented. Are these “digital health experts” already part of or members of the hospitals and/or sponsors, or are they meant to be a different entity completely?